# Neurobehavioral Outcomes of Mild Traumatic Brain Injury: A Mini Review

**DOI:** 10.3390/brainsci7050046

**Published:** 2017-04-25

**Authors:** Robert Eme

**Affiliations:** Illinois School of Professional Psychology, Argosy University, Schaumburg, IL 60202, USA; reme@argosy.edu

**Keywords:** traumatic brain injury, post-concussive syndrome, diffuse axonal injury

## Abstract

Traumatic brain injury outcomes can be classified as acute or chronic. Acute outcomes refer to injuries that occur immediately at the time of the injury and subsequent short-term consequences. Chronic outcomes refer to adverse outcomes that are more long-term. In mild traumatic brain injury, recovery from acute outcomes typically occurs very rapidly, i.e., within 2 weeks, with full recovery expected by 90 days. However, some 10%–15% individuals can remain symptomatic for much longer with an outcome termed post-concussive syndrome. This outcome is difficult to predict since there are very few rigorous, prospective studies of this syndrome.

## 1. Introduction

Interest in understanding the outcomes of mild traumatic brain injury (mTBI), also known as concussion (these terms will be used interchangeably in the review), has markedly increased in the past decade as the general public has become more aware of the rates and consequences of mTBI, especially in athletic and military settings [1,2]. This increased awareness and concern about the consequences of mTBI warrant a concise review of the outcomes of mTBI, especially since it has become increasingly clear that many cases of mTBI can cause persistent neurobehavioral impairment [1,2]. This article will provide such a review, beginning with a discussion of TBI that will provide the foundation for a definition of mTBI.

## 2. Traumatic Brain Injury

Traumatic brain injury (TBI) is a traumatically induced structural injury or other alteration in brain function as a result of an external force resulting in a new onset or worsening of at least one of the following clinical signs immediately following the event [3,4]:any period of loss or of a decreased level of consciousness;any loss of memory for events immediately before or after injury;any alteration in mental state at the time of injury (confusion, disorientation, slowed thinking, etc.);neurological deficits (weakness, loss of balance, change in vision, etc.).

TBI can include penetrating injuries (in which the object penetrates the skull and dura, with direct damage to the brain) and closed-head injuries in which the skull and dura remain intact [4]. TBI can be categorized into mild, moderate, and severe based on clinical factors, such as the duration of loss of consciousness (if present), presence of amnesia, neurological symptoms, and the results of structural brain imaging, typically computed tomography (CT) and magnetic resonance imaging (MRI) [4]. Initial signs of neurological symptoms are typically assessed for by scores on the Glasgow coma scale (GCS), which is a simple measure of best speech and language, motor, and oculomotor function within 24 h of injury [5]. On this scale, which historically has been considered the gold standard in the assessment of initial injury severity, individuals are given a total score ranging from 1 to 15 based on degree of impairment of the three functions, with lower scores indicating more impairment. By convention, scores from 13 to 15 represent mild injuries, scores from 9 to 12 represent moderate injuries, and scores of 8 or less represent severe injuries [4,5,6,7,8].

## 3. Mild Traumatic Brain Injury

Despite some inconsistency among researchers, mTBI (which represents 80%–90% of cases) is typically defined by the following criteria [4,7]:normal structural imaging;loss of consciousness for 0–30 min;altered mental state duration for <24 h;post-trauma amnesia for <1 day;GCS 13–15.

However, the assessment procedures employed to classify individual as mild have two major limitations that help explain, as will be subsequently discussed, why some putatively mild cases of TBI can have chronic outcomes. First, the neuroimaging techniques of CT and MRI that are typically used to assess the pathophysiology of mTBI do not have sufficient sensitivity to detect mild multifocal axonal injury, commonly referred to as diffuse axonal injury (DAI)—the most important mechanism for persistent neuronal injury following TBI [1,2,4,9,10,11,12]. DAI is caused by the acceleration and deceleration of three kinds of mechanical forces that result in strain and shearing on the axons of the brain: (a) linear translation occurs as a result of forces that make the head move in an anterior–posterior direction (such as hitting the front or back of the head); (b) rotational acceleration occurs as a result of forces that make the head move sideways (such as a punch to one side of the head; (c) impact deceleration occurs when the head forcefully decelerates (such as when the head hits the ground. The second limitation is that, in general, the GCS is considered a crude tool for assessing “…one of the most complex heterogeneous disorders in the most complex organ in the body and dumbing it down to mild, moderate, and severe” [12] Indeed, several studies have shown the development of traumatic intracranial brain hematomas in 15%–20% of individuals with a perfect GCS score of 15. For example, Kirkwood and Yeates presented a case history of a 15-year-old male with an mTBI who, despite a perfect GCS score of 15, had a small right frontal hematoma, and at 6 weeks post-injury experienced impaired functioning on a neuropsychological evaluation, and severe academic problems [12]. In addition, the GCS has never been fully validated for use with children [6,8].

In conclusion, since mTBI is defined on clinical grounds and typically evaluated using imaging techniques unable to detect DAI, many cases of TBI classified as mild can be more severe and hence more likely to result in more chronic negative neuropsychological outcomes, as will be subsequently discussed.

## 4. Outcomes

TBI outcomes can be classified as acute or chronic [13] Acute outcomes refer to injuries that occur immediately at the time of the TBI and subsequent short-term consequences. Chronic outcomes refer to adverse outcomes that are more long-term.

## 5. Acute Outcomes

The acute outcomes of a concussion affect three domains: cognitive, behavioral, and physical [14]. Common cognitive outcomes are decreased speed of information processing, attention problems, and confusion. Common behavioral outcomes are irritability, emotional lability, and hyperactivity. Common physical outcomes are headache, dizziness, and nausea. Of all these outcomes, attention problems and hyperactivity are the most common [15]. Recovery from these outcomes typically occurs very rapidly, i.e., within 2 weeks, with full recovery expected by 90 days [2,4,15]. However, some 10%–15% individuals can remain symptomatic for much longer [2,4,15].

## 6. Chronic Outcomes

The persistence of symptoms such as poor concentration, dizziness, fatigue, headache, sleep disturbance, irritability, anxiety, and depressed mood beyond 30 days is termed post-concussive syndrome (PCS). This more chronic outcome is due to cases at the extreme end of the mild range of TBI being most at risk of persistent adverse neuropsychological outcomes [12]. The term complicated was coined in 1990 to designate these cases at the severe end of mPTBI when it was discovered that the presence of abnormal CT findings in mild cases of TBI revealed complications of brain physiology and predicted more persistent problems [16]. Since then, research that supports this initial finding has been accumulating [4]. The most compelling evidence comes from research using more sophisticated neuroimaging techniques such as diffusion tensor imaging (DTI) which, unlike CT and MRI, can detect DAI. (CT is commonly used in mTBI cases since it is fast and can accurately detect all life-threatening and surgically treatable intracranial hemorrhages in a patient with TBI [4]. MRI can provide more sensitive detection of smaller hemorrhages than CT. However, it is less clinically available and has longer scan times that CT [4].) DTI maps axonal organization by measuring the diffusion of water molecules in the brain, with a reduction in diffusion providing evidence of axonal injury [4]. For example, in a study of 63 soldiers who had been diagnosed with an mTBI, 29% showed DTI abnormalities, but no abnormalities on an MRI scan [11]. Similarly, in addition to DTI, another new neuroimaging technique, gradient-echo MRI, has been developed that was able to detect microhemorrhages indicative of DAI in an athlete who had sustained a concussion [15].

An mTBI is more likely to have a PCS outcome when it after occurs after one or more prior concussions with a deficit in executive functions appearing to be the most common cognitive deficit [2]. (Although definitions of executive functions vary considerably, a generally accepted broad definition of executive functions is that it refers to cognitive processes that are required for top-down control of behavior, emotions and thoughts that are associated with neural systems involving the prefrontal cortex [17]. These outcomes are especially likely to ensue when a subsequent mTBI occurs soon after the prior mTBI, and therefore before the resolution of the pathophysiological changes following the prior concussion [2,4,15]. After an mTBI, the brain has an increased vulnerability for a new injury during the recovery phase termed the “window of vulnerability” [4]. During this “window of vulnerability,” a re-injury can add to the cumulative damage to the brain and thus prolong recovery from this new injury. Furthermore, there is now convincing evidence that repeated concussive or subconcussive injuries are linked to a neurodegenerative condition termed chronic traumatic encephalopathy (CTE) [18]. (As it is beyond the scope of this article to conduct a review of CTE, the interested reader is advised to consult the 2015 review by Riley and colleagues which is the most current, comprehensive review known to the author [18].

In summary, an mTBI can result in the chronic neuropsychological outcome of PCS, especially when it occurs soon after a prior mTBI and/or is yet another instance of multiple prior mTBIs. Unfortunately, PCS is difficult to diagnose or predict as no objective diagnostic tools are available and there is little evidence to identify a threshold when multiple concussions could precipitate either PCS or CTE [4]. It is likely that genetic factors contribute to a chronic outcome through their influence on the severity of axonal injury [2]. In addition, various personality, psychiatric, and environmental factors can also affect symptom severity and chronicity [2,4,15]. For example, one of the most robust non-neural predictors of outcome is pre-injury functioning in that individuals with higher levels of behavior and social problems tend to have more persistent problems [2].

## 7. Conclusions

Most patients with mTBI will recover quickly, but approximately 10%–15% will have a more chronic course called PCS. Unfortunately, since there are very few rigorous, prospective studies of PCS, this outcome is difficult to predict [15]. Despite this disheartening lack of research, there is still a very important implication for clinical practice. It is the recognition that a TBI that, from all clinical criteria, is viewed as mild can indeed have a more chronic outcome that is probably due to DAI. This recognition should prevent the clinician from erroneously concluding that the persistent symptomatology following an mTBI is necessarily a product of various non-neural, environmental factors or perhaps malingering.

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
