# Peer review of "Neurobehavioral Outcomes of Mild Traumatic Brain Injury: A Mini Review"

_brainsci, 2017, doi:10.3390/brainsci7050046_

Round 1

Reviewer 1 Report

This article calls itself "A Mini Review" and that's what it is, not more and not less. Traumatic Brain Injuries (TBI) is a hot topic and, as stated in the paper, their outcome and long-term prognosis may be difficult. As it is the case with reviews there is nothing new or novel to it.

This is a well-structured and well written short article and I recommend publication in its present form. 

Author Response

Please check the attached version.

Reviewer 2 Report

The author briefly introduced background of TBI and reviewed outcomes of mild TBI. The major topic of this mini review is persist neurobehavioral impairment can be the outcome of mild TBI.  This topic is narrow enough for mini review. However the scope of title seems so wide for this topic. The author may consider to modify the title.  This manuscript is well written except the sentences from line 95 to line 99 are difficult to understand.

In line 53, number of day is missing.

Author Response

Please check the attached version.
